

# Wigner function analysis of finite matter-radiation systems

Eduardo Nahmad-Achar⋆, Ramón López-Peña,
Sergio Cordero and Octavio Castaños Garza

Instituto de Ciencias Nucleares, Universidad Nacional Autónoma de México,
Apartado Postal 70-543, México 04510 CDMX

⋆ nahmad@nucleares.unam.mx

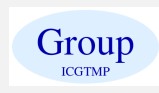

## Abstract

**We show that the behaviour in phase space of the Wigner function associated to the electromagnetic modes carries the information of both, the entanglement properties between matter and field, and the regions in parameter space where quantum phase transitions take place. A finer classification for the continuous phase transitions is obtained through the computation of the surface of minimum fidelity.**

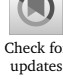

## 1 Introduction

Quantum phase transitions (QPT) are studied in nuclear, molecular, quantum optics, and condensed matter physics, and have potential applications in the design of quantum technologies [1]. The Wigner function gives a complete description of a quantum system in phase space; it allows for the calculation of all the quantities that the usual wave function gives, and negative values in the function appear as a consequence of interference between distant points in phase space. In a generalized Dicke model of 3-level atoms interacting with 2 electromagnetic modes, it may be used to analyse the behaviour in phase space of the two radiation modes of light across the finite phase diagram of the quantum ground state, and supply further evidence of the quantum phase transitions revealed by the fidelity criterion.

When the linear entropy for all the subsystems is calculated and compared with the behaviour of the Wigner function, we see that the entanglement between the substates responds to how the bulk of the ground state changes from a subset of the basis with a major contribution from one kind of photons, to a subset with a major contribution of the other one.

## 2  The generalised Dicke model

The multipolar Hamiltonian for the dipole interaction between a 2-mode radiation field and a 3-level atomic system in the long wave approximation ($\hbar = 1$) is

$$\mathbf{H} = \mathbf{H}_D + \mathbf{H}_{int}\,,$$

with

$$\mathbf{H}_D = \sum_{j<k}^{3} \Omega_{jk}\, \mathbf{a}_{jk}^{\dagger}\, \mathbf{a}_{jk} + \sum_{j=1}^{3} \omega_j\, \mathbf{A}_{jj}\,,$$

and

$$\mathbf{H}_{int} = -\frac{1}{\sqrt{N_a}} \sum_{j<k}^{3} \mu_{jk}\left(\mathbf{A}_{jk} + \mathbf{A}_{kj}\right)\left(\mathbf{a}_{jk} + \mathbf{a}_{jk}^{\dagger}\right).$$

Here, $N_a$ denotes the number of particles, $\mathbf{a}_{jk}^{\dagger}$, $\mathbf{a}_{jk}$ are creation and annihilation photon operators, $\Omega_{jk}$ is the frequency of the mode which promotes transitions between the atomic levels $\omega_j$ and $\omega_k$, $\mathbf{A}_{ij}$ are the matter operators obeying the $U(3)$ algebra, with $\sum_{k=1}^{3} \mathbf{A}_{kk} = N_a\, \mathbf{I}_{\text{matter}}$, and $\mu_{jk}$ is the coupling parameter between atomic levels $\omega_j$ and $\omega_k$.

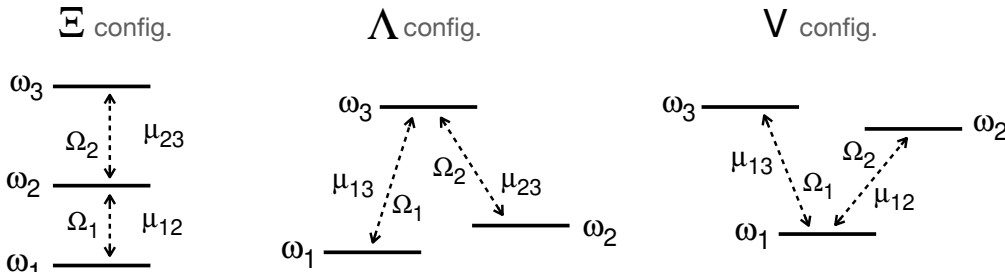

Figure 1: Atomic configurations for 3-level systems, showing the possible transitions and coupling strengths $\mu_{ij}$.

We have the atomic configurations shown in Figure 1, customarily labelled by $\Xi$, $\Lambda$, and $V$, due to their shape resembling these letters, and where we label the atomic energy levels following $\omega_1 \leq \omega_2 \leq \omega_3$ and for simplicity fix $\omega_1 = 0$ and $\omega_3 = 1$; therefore, all energies are measured in terms of $\hbar\omega_3$. Note that particular atomic configurations are obtained by making an appropriate dipolar strength $\mu_{ij}$ vanish.

### 2.1  Variational study

A variational study involving coherent states for both matter and field provides a good approximation of the ground state energy surface per particle [2, 3]. Figure 2 shows the phase diagrams from a variational study using coherent test states, for the different atomic configurations $\Xi$, $\Lambda$, and $V$ (from left to right), as well as the order of the transitions according to the Ehrenfest classification. We distinguish a *normal* region ($N$, in medium grey) where the atoms decay individually, and *collective* regions $S_{ij}$ where the decay is proportional to $N_a(N_a + 1)$ and in which only one kind of photon contributes to the ground state. Continuous black lines denote the separatrices dividing these regions.

It is important to note that the signature of the phase diagram remains when the symmetries of the Hamiltonian are restored in the variational solution and the thermodynamic limit $N_a \to \infty$ is taken.

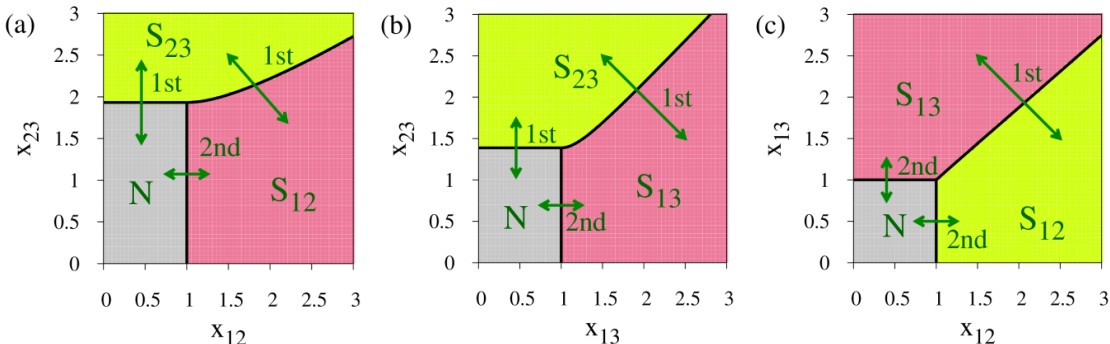

Figure 2: Phase diagrams from a variational study using coherent test states, for the different atomic configurations $\Xi$, $\Lambda$, and $V$ (from left to right). The order of the transitions according to the Ehrenfest classification is shown. The parameters used are: $\Xi$-configuration: $\omega_2/\omega_3 = 1/3$; $\Lambda$- configuration: $\omega_2/\omega_3 = 1/10$; $V$-configuration: $\omega_2/\omega_3 = 8/10$. Here, $x_{ij} = \mu_{ij}/\mu_c$ is the dimensionless dipolar coupling strength, where $\mu_c$ stands for its critical value in the two-level system $\{ij\}$ in the limit $N_a \to \infty$.

## 2.2 Numerical quantum solution

The exact calculation of the ground state involves a numerical diagonalisation of the Hamiltonian matrix. The Hamiltonian is invariant under parity transformations of the form

$$\Pi_1 = e^{i\pi \mathbf{K}_1}, \quad \Pi_2 = e^{i\pi \mathbf{K}_2},$$

where $\mathbf{K}_s$, $s = 1, 2$, are constants of motion when the rotating wave approximation (RWA) is taken [7]. Accordingly, the Hilbert space $\mathcal{H}$ divides naturally into four subspaces

$$\mathcal{H} = \mathcal{H}_{ee} \oplus \mathcal{H}_{eo} \oplus \mathcal{H}_{oe} \oplus \mathcal{H}_{oo},$$

where subscripts $\sigma = \{ee, eo, oe, oo\}$ denote the even $e$ or odd $o$ parity of $\Pi_1$ and $\Pi_2$, respectively.

We use basis states labeled by $|\nu_{12}, \nu_{13}, \nu_{23}\rangle \otimes |n_1, n_2, n_3\rangle$, with $n_1 + n_2 + n_3 = N_a$ and $\nu_{jk} = 0, 1, \cdots, \infty$, which denote Fock states.

Since the dimension of the Hilbert space is $\dim(\mathcal{H}) = \infty$, we need to use a truncation criterion. For the set of eigenvalues of $\mathbf{K}_1$, $\mathbf{K}_2$, we take this criterion as follows [8]: choose values $k_{i\max}$ to satisfy

$$1 - \mathcal{F}(k_{1\max}, k_{2\max}) \leq 10^{-10},$$

where $\mathcal{F}(k_1, k_2) = |\langle \psi(k_1, k_2) | \psi(k_1 + 2, k_2 + 2)\rangle|^2$ is the fidelity between the state $|\psi(k_1, k_2)\rangle$ containing all eigenvalues up to $k_1$ and $k_2$, and the state $|\psi(k_1 + 2, k_2 + 2)\rangle$. This ensures that the energy calculated remains without variation to one part in $10^{-8}$. Other criteria may be used, of course, depending on the problem in question.

## 3 Fidelity as signature of QPT in *finite* systems

Quantum phase transitions are determined by singularities in the wave function of the ground state, and these may be studied by the method of Ginzburg-Landau, or using catastrophe theory, in the thermodynamic limit [4]. Another criterion is by the loci where the fidelity between neighbouring states $|\Psi_g(\xi_1)\rangle$, $|\Psi_g(\xi_2)\rangle$ along parametric lines $\xi(t)$ in parameter space

$$\mathcal{F}(\rho_{\xi(t)}, \rho_{\xi(t+\delta)}) = |\langle \Psi_g(\xi(t)) | \Psi_g(\xi(t+\delta))\rangle|^2,$$

presents a minimum (see e.g. [5,6] and references therein). We have followed this method for *finite* systems to find the separatrices in parameter space. We call these *quantum phase transitions*, in contrast to other terminology that appears in the literature, since the constitution of the ground state changes significantly as one crosses a separatrix. The *surface of minimum fidelity* is calculated by considering neighbouring points in directions parallel to the axes ($x_{jk} = 0$), along identity lines, and along their orthogonal directions, thereby finding the local minima. Here, $x_{jk} = \mu_{jk}/\mu_c$, where

$$\mu_c = \frac{1}{2}\sqrt{\Omega_{jk}(\omega_k - \omega_j)},$$

stands for the critical coupling value in a two-level $\{jk\}$ system, in the limit $N_a \to \infty$. Thus, $x_{jk}$ is the dimensionless dipolar coupling.

In the case of the generalised *quantum* Rabi model, the quantum separatrices for a single 3-level atom interacting dipolarly with two modes of electromagnetic field are given in Figure 3, for the three atomic configurations, $\Xi$, $\Lambda$, and $V$ (from left to right), when in resonance with the field modes [7]. The parity of the Hilbert subspace in which the ground state lives is marked by colours and by the letters $\{ee, eo, oe, oo\}$, and we see that a much richer structure appears in contrast with the limit $N_a \to \infty$ shown in Fig. 2.

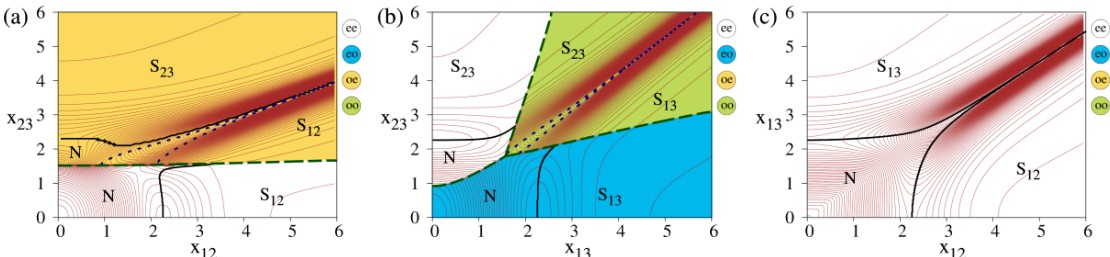

Figure 3: Quantum phase diagrams for the three atomic configurations, $\Xi$, $\Lambda$, and $V$ (from left to right), for one atom when in resonance with the field modes. Different types of transitions are shown (see text). For the $\Xi$-configuration we have used $\Omega_{12} = 1/4$, $\Omega_{23} = 3/4$ and $\omega_2 = 1/4$; for the $\Lambda$-configuration $\Omega_{13} = 1$, $\Omega_{23} = 9/10$, and $\omega_2 = 1/10$; and for the $V$-configuration $\Omega_{12} = 4/5$, $\Omega_{13} = 1$ and $\omega_2 = 4/5$.

Quantum phase transitions for a finite system appear where the ground state changes abruptly, and this may be determined by calculating the fidelity or the fidelity susceptibility between neighbouring states. We can distinguish three types of loci of points where this takes place (cf. Figure 3):

1. Dashed lines: *discontinuous transitions*, the fidelity between neighbouring states falls to zero, and the separatrix in this case borders along orthogonal Hilbert subspaces of different parity;

2. Continuous lines: *stable continuous transitions*, $F(\xi) \neq 0$ and it remains different from zero as $N_a$ increases;

3. Dotted lines: *unstable continuous transitions*, $F(\xi) \neq 0$ but reaches zero in the large $N_a$ limit.

This classification is further corroborated through the behaviour of the Wigner function for each field mode, as we shall see in the next section. Note that stable and unstable continuous transitions can also be distinguished by means of the Bures distance, which measures the difference of two probability densities of the quantum system; for the stable continuous transition the value of the Bures distance will be smaller than for the unstable continuous transition.

# 4 Wigner function in the Λ-configuration

First order quantum phase transitions, according to the Ehrenfest classification, can be always associated to zero fidelity values, i.e., discontinuous transitions, and the corresponding eigenstates are orthogonal.

A finer classification of the continuous transitions is more evident through the study of the *Wigner function*, since this classification is based on whether the bulk of the ground state remains in a sub-basis of the total basis or not. Here we shall focus on the Λ-configuration, which appears to have a richer structure.

We may use the parity operators for the Λ-configuration

$$\mathbf{K}_1 = \boldsymbol{\nu}_{13} + \boldsymbol{\nu}_{23} + \mathbf{A}_{33},$$
$$\mathbf{K}_2 = \boldsymbol{\nu}_{23} + \mathbf{A}_{11} + \mathbf{A}_{33},$$

to replace the electromagnetic quanta oscillation numbers

$$\nu_{13} = k_1 - k_2 + n_1, \qquad \nu_{23} = k_2 - n_1 - n_3,$$

and thus denote the ground state of the system as

$$|\psi_{\text{gs}}\rangle = \sum_{k_1,k_2} \sum_{n_1,n_3}^{N_a} C_{k_1,k_2,n_1,n_3} \times |k_1 - k_2 + n_1, k_2 - n_1 - n_3, n_1, N_a - n_1 - n_3, n_3\rangle,$$

from which we calculate the reduced density matrices $\varrho_{jk}$ ($j < k$) for modes $\nu_{jk}$.

Notice that for the case of a single atom, for maximum values of $x_{jk} = 6$ and for the desired precision of $10^{-10}$ established in Sec. 2.2, the ground state function lives in a Hilbert space of dimension $\dim(\mathcal{H}) = 1395$, while for a precision of $10^{-15}$ the dimension must at least be $\dim(\mathcal{H}) = 2079$ [8].

Thus, the Wigner functions for the reduced density matrices are

$$W_{13}(q,p) = \sum_{k_1,k_2,k_1'} \sum_{n_1,n_3} C_{k_1,k_2,n_1,n_3} C^*_{k_1',k_2,n_1,n_3} W_{|k_1-k_2+n_1\rangle\langle k_1'-k_2+n_1|}(q,p),$$
$$W_{23}(q,p) = \sum_{k_1,k_2,k_2'} \sum_{n_1,n_3} C_{k_1,k_2,n_1,n_3} C^*_{k_1,k_2',n_1,n_3} W_{|k_2-n_1-n_3\rangle\langle k_2'-n_1-n_3|}(q,p),$$

where $W_{|n\rangle\langle m|}(q,p)$ is the Weyl symbol for the operator $\rho_{nm} = |n\rangle\langle m|$ [9,10].

We may plot these Wigner functions as functions of the field quadratures $(q,p)$ at various points at either side of a separatrix, to see their behaviour as the system undergoes a phase transition [7].

Figure 4 shows the behaviour of $W_{13}$ as the system goes through a stable-continuous transition (red dot along a continuous grey evaluation trajectory). The elongation presenting a bimodal distribution is a consequence of photon contribution $\nu_{13}$ becoming significant. Regions where the Wigner function $W_{13}$ is negative (black) appear as we move away from the normal region and cross the separatrix, because the number of photons in mode $\nu_{13}$ grows from zero: we now have a superposition of states with different values of $\nu_{13}$.

Figure 5 shows the behaviour of both, $W_{13}$ and $W_{23}$, as the system goes through an unstable-continuous transition (red dot along a continuous grey evaluation trajectory): close to the separatrix in dotted lines both photon contributions are significant. We note that both Wigner functions present elongated (bimodal) distributions. Above the separatrix the contribution of photons $\nu_{23}$ dominates and $W_{23}$ has major regions with negative values; when the transition occurs, the field mode contributions to the ground state change their roles.

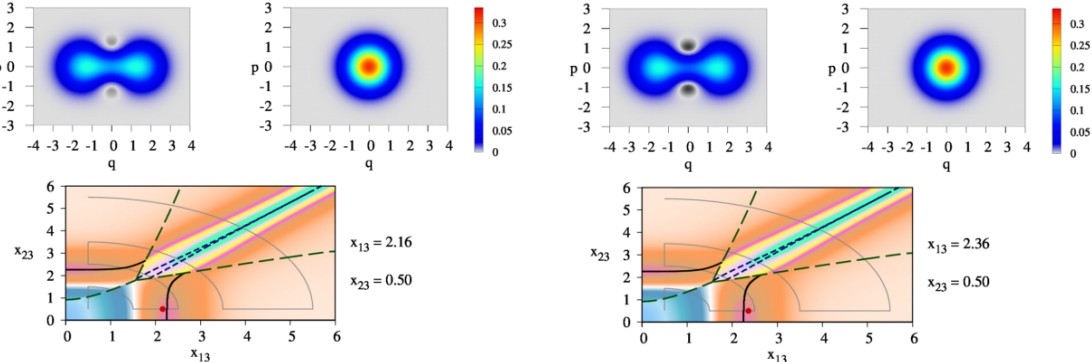

Figure 4: Behaviour of the Wigner function $W_{13}$ and $W_{23}$, as the system goes through a stable-continuous transition. Regions where it becomes negative (black) reflect the existence of a superposition of states with different values of $\nu_{13}$. (In each case, the continuous dim grey line is the evaluation trajectory, the red dot indicates the evaluation point in parameter space.) Note that, through this transition, $W_{23}$ does not change.

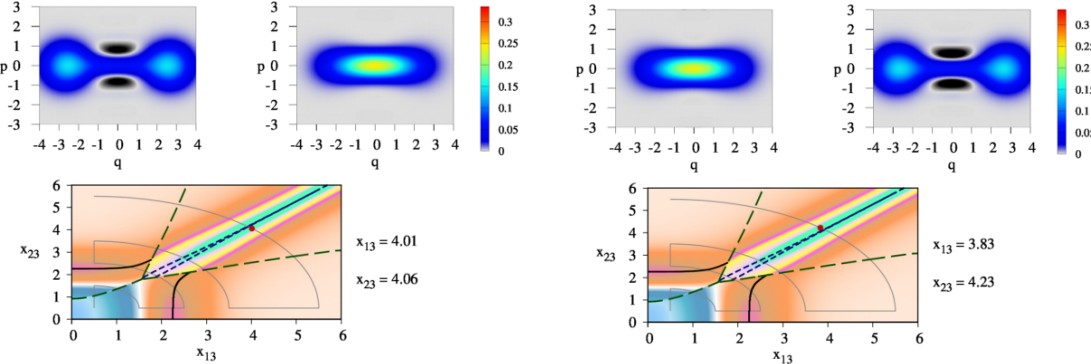

Figure 5: Behaviour of $W_{13}$ and $W_{23}$ as the system goes through an unstable-continuous transition. Across the transition the field mode contributions to the ground state change their roles $S_{13} \rightleftharpoons S_{23}$. (In each case, the continuous dim grey line is the evaluation trajectory, the red dot indicates the evaluation point in parameter space.)

We see that the Wigner function characterises completely the phase diagram. In the normal region the Wigner function describes a classical behaviour of the field ($W$ takes positive values) and at least one photon mode remains in the vacuum, while the collective region is characterised by a Wigner function in which the quantumness of the photon modes is clearly shown; it divides itself into two regions, in each of which a single radiation mode dominates.

Videos showing the behaviour of the Wigner function for each mode, along the full trajectory shown in Figure 5, may be found for all the atomic configurations in the website of IOP *Physica Scripta*: $\Xi$-configuration; $\Lambda$-configuration; $V$-configuration.

## 4.1 Correlation between Wigner function and entanglement

Bimodality and negativity of Wigner function reflect which field mode dominates in the superradiant region, and not the parity of the state. This is evident when we compare it with an

entanglement measure (e.g., the linear entropy) [7]. We define

$$S_{\nu_1} = 1 - \text{Tr}\left(\rho_{\nu_1}^2\right),$$

$$S_{\nu_2} = 1 - \text{Tr}\left(\rho_{\nu_2}^2\right),$$

$$S_{\nu-m} = 1 - \text{Tr}\left(\rho_{\nu_1\nu_2}^2\right),$$

to be, respectively, the linear entropy measuring correlation between field mode 1 and the rest of the system (matter + field mode 2), the linear entropy measuring correlation between field mode 2 and the rest of the system (matter + field mode 1), and the linear entropy measuring correlation between matter and field modes 1 and 2.

Figure 6 shows their plots along a trajectory which crosses all detected transitions in parameter space. When the ground state is dominated by the vacuum state of the field (small values of the coupling parameters inside the Normal region), the correlation between one mode of the field, say $i$, and the rest of the system (matter + field mode $j$ with $i \neq j$), is null $S_{L_i} = 0$ and the Wigner function is unimodal. This field-mode $i$ vs. matter + field-mode $j$ entanglement reaches its maximum as soon as we cross into the super-radiant region, the Wigner function showing negative values at a vicinity of the origin of quadrature $q$ and small non-zero values of quadrature $p$. It then falls rapidly to zero as soon as we enter the region where field mode $j$ dominates, even if a parity change is not had.

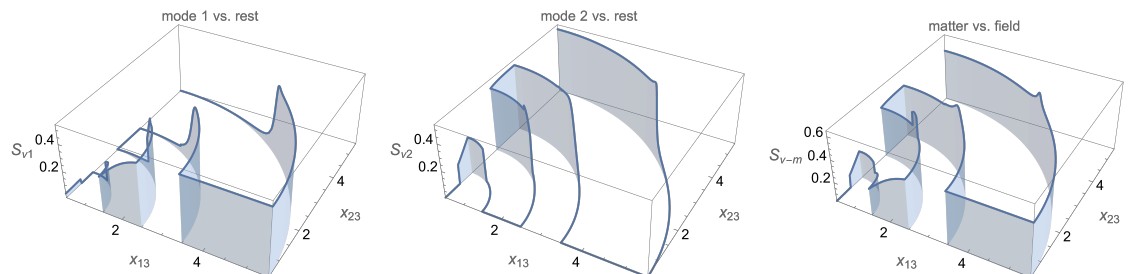

Figure 6: Plots of the different linear entropies $S_{\nu 1}$, $S_{\nu 2}$, and $S_{\nu-m}$, along a trajectory which crosses all detected transitions in parameter space.

## 5 Conclusion

We have shown the results of the characteristics of the ground state for a single three-level atom interacting dipolarly with a two-mode electromagnetic field. The symmetries of the system allow for the division the quantum state space into subspaces which have a well-defined parity. We have used a fidelity criterion to determine the quantum phase transitions for the three three-level configurations.

We calculated the Wigner function for each of the electromagnetic modes $\Omega_{13}$ and $\Omega_{23}$, and showed the behaviour of these in various regions of the parameter space, which supplies further evidence of the quantum phase transitions revealed by the fidelity criterion; the regions where it takes negative values (the system exhibiting non-classical behaviour) were determined. Besides providing the phase transitions and a finer classification of them, it is interesting to note that the Wigner function can be and has been measured experimentally [11, 12].

The linear entropy for all the subsystems was calculated and compared with the behaviour of the Wigner function; we see that the entanglement between the substates responds to how the bulk of the ground state changes from a subset of the basis with a major contribution from

one kind of photons, to a subset with a major contribution of the other one, and not to the state parity even for large values of the coupling parameters.

## 6 Acknowledgements

**Funding information** This work was partially supported by DGAPA-UNAM (under projects IN112520, and IN100323).

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
