# Peer review of "Wigner Function Analysis of Finite Matter-Radiation Systems"

_SciPost Physics Proceedings, doi:SciPost Phys. Proc. 14, 039 (2023)_

## Round 1 · Referee Report · Anonymous · 2023-2-6

Report
This contribution is interesting an well-written, thus I think that
it should be published. There are only some minor changes and suggestions that should be addressed
Requested changes
- Explain the meaning of $x_{12}$ and $x_{23} $ in Fig. 2 (its definition is not given until Sec. 3).
- Change the notation for the maximum number of photons $k_i$ of mode $i$ at the beginning of Sec. 2, since this notation creates confusion with the eigenvalues $k_i$ of the parity operators in Sec. 4.
- In Fig. 4 two plots of Wigner functions appear in each case (red dots), but in the caption (and in the main text) reference is made to just $W_{13}$. Comment the behavior of the other Wigner function, $W_{23}$ (even if it doesn't change), or delete it from the Figure.
Anonymous on 2023-01-25 [id 3273]
In this submission the authors discuss phase-transitions in three-level Dicke model by analyzing the fidelity and the Wigner function of field mode. The presentation is clear and publishable. I believe that authors should consider the following comments: 1. It would be better if Sec. 2 were titled “Numerical solution". Usually, the term “exact solution" is understood to be an analytical solution in terms of elementary functions. 2. Both the fidelity and the continuous phase-space distributions have been widely used for detection of phase transitions. Please cite appropriate papers. 3. There is no well-defined criterion for the appearance of phase transitions in the manuscript: is it the negativity or the splitting of the distribution (both criteria were used before in the literature) Please, define such a criterion. 4. I do not think that the Eqs. after the line 113 are really needed, since they are direct consequence of the expression for the state vector |psi_gs>.

---

## Round 2 · Author Response

All recommendations from both, the Report and the Comment, have been taken into account.
We thank both reviewers for their careful review and assertive comments.

---

## Round 2 · List of Changes

Report:

1) x_jk has been explicitly described both, in the caption of Figure 2, and in the text in the beginning of Section 3.

2) k_i has been explained more carefully, as well as the truncation criterion and the calculation of the fidelity between states.

3) The caption of Figure 4 has been expanded for greater clarity.

Comment:

1) Title of Subsection 2.2 has been changed to “Numerical Quantum Solution”.

2) References have been added for the use of the fidelity criterion in determining quantum phase transitions.

3) A well defined criterion is now given at the end of p.4.

4) The equations on (previous) line 113 have been eliminated.

Others:

Three new references were added.

---

## Editorial Decision

published